# MULTI-FIDELITY FINE-TUNING OF PRE-TRAINED LANGUAGE MODELS

## ABSTRACT

We consider the problem of fine-tuning pre-trained language models with a small amount of trusted data (high-fidelity) and a larger amount of data with noisy labels (low-fidelity). We propose Multi-Fidelity Fine-Tuning (MFFT), a novel approach which implicitly determines for new inputs when we can rely on information from high-fidelity data and when instead we need to fall back on knowledge from low-fidelity data. MFFT does not require any architecture changes to the base model and simply provides its fine-tuned version that can be easily deployed for inference. We extensively benchmark MFFT on various classification tasks against several baselines, with both simulated label noise, and in realistic scenarios with LLM generated data. MFFT consistently improves performance compared to using trusted data alone and outperforms all baselines across experiments with macro F1-score improvements of 2-4%. Finally, it provides substantial improvements in uncertainty calibration with expected calibration error (ECE) reductions of 40-60% compared to the best baselines.[1]

## 1 INTRODUCTION

Fine-tuning foundation models (Bommasani et al., 2021) using a small amount of clean data and more abundant noisy data is ubiquitous to many deep learning settings, such as learning from expert and non-expert annotated data (Shapiro et al., 2013; Su et al., 2012; Sylolypavan et al., 2023), training end models with weak supervision (Zhang et al., 2022; Ratner et al., 2017; Rühling Cachay et al., 2021) and learning from humans and large language models (LLMs) in combination (Thapa et al., 2023; Ding et al., 2024; Meng et al., 2023; Zhang et al., 2024; Wang et al., 2023a; Li et al., 2021b). However, combining data sources of varying quality for fine-tuning effectively is not trivial. As shown in Figure 1, simply adding lower quality data to the training set is often detrimental to model performance (Zhou et al., 2024; Wang et al., 2023a). While training neural networks by combining noisy and clean data has been extensively studied (Song et al., 2022; Hendrycks et al., 2018; Patrini et al., 2017; Veit et al., 2017), how to develop a strategy to fine-tune pre-trained language models (Vaswani et al., 2017; Radford et al., 2019; Dubey et al., 2024) has yet to be explored.

We propose Multi-Fidelity Fine-Tuning (MFFT), a novel language model fine-tuning approach for down-stream tasks which leverages small amounts of data with trusted labels in combination with a larger data set with noisy labels. Our approach uses two fine-tuned versions of the base model to infer pseudo-labels that are used to fine-tune a final model (Figure 2). We first fine-tune a *low-fidelity* model using the abundant noisy data. Then, we fine-tune a *high-fidelity model* with the scarce clean data. Based on the expected log likelihood, MFFT determines whether the low or high fidelity model should be used for inference given a new input. This selective inference is run over the whole low-fidelity data set to infer soft labels. The pseudo-labeled version of the original low-fidelity data set is used together with the high-fidelity data to fine-tune the base model resulting into a final model. This final fine-tuned model implicitly learns when it can infer using information gained from high-fidelity data and when, conversely, it has to fall back on predictions learned from low-fidelity data, which are less accurate, due to the noise in the labels, but more robust, due to the abundance of examples.

In our extensive evaluation, we use MFFT to fine-tune different language models for a variety of classification tasks with a small amount (50-100 examples) of trusted data and a larger noisy data

---

[1]Code will be made publicly available.

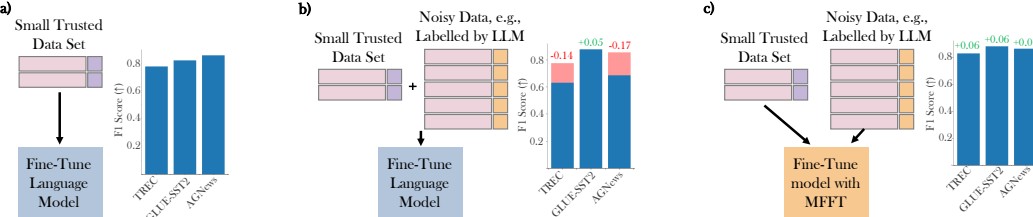

Figure 1: Fine-tuning language models with mixed quality data. **a)** As reference, we can fine-tune the model with the small amount of high-quality trusted data only. **b)** adding large amounts of lower quality data, e.g., labeled or generated with an LLM, to the fine-tuning set can actually degrade performance, instead of improving it. **c)** Our MFFT method incorporates lower quality data with awareness of its lower reliability, resulting in fine-tuned models that consistently benefit from it.

set ($\sim 5,000$ examples). We experiment by using simulated noisy low-fidelity data, as well as in a more realistic setting where this data is obtained by generating or annotating examples with an LLM. Models fine-tuned with MFFT consistently obtain competitive or better performance compared to using trusted data only (example in Figure 1) across all experiments. MFFT also performs competitively or better than all tested baselines across different models, noise properties and tasks, with improvements in macro F1-score of 2-4% compared to the best baseline and reductions in expected calibration error (ECE) of 40-60%, which indicates a large improvement in uncertainty calibration.

## 2 BACKGROUND AND RELATED WORK

### 2.1 LEARNING WITH NOISY AND CLEAN LABELS

The problem of learning with noisy labels (LNL) has been extensively studied (Song et al., 2022) for exploiting a small amount of data with clean labels in combination with larger amounts of data with noisy labels. In the context of LNL, available clean data is often referred to as trusted data or anchor points (Song et al., 2022; Patrini et al., 2017). Some methods propose to use the anchor points to first learn a label correction model for inferring a clean label jointly from inputs and noisy label (Xiao et al., 2015; Zheng et al., 2021; Veit et al., 2017). Other methods propose instead to use the clean data to design and calibrate a noise-robust cost function (Hendrycks et al., 2018; Patrini et al., 2017). While proven effective in deep learning, these methods are difficult to apply directly to fine-tuning foundational models. Firstly, because of the data regime we target; As fine-tuning pre-trained language models requires much less data than training neural networks from scratch (Zhou et al., 2024; Qiu et al., 2020), we aim to push the boundaries of learning from mixed quality data in terms of clean data requirements and use only up to tens of clean examples per class. This causes label correction and cost calibration methods to over-fit. Secondly, many existing approaches require specific model architectures (Song et al., 2022; Zheng et al., 2021), while we wish to maintain our fine-tuning strategy applicable in a plug-and-play fashion to any foundation model.

### 2.2 MULTI-FIDELITY MODELS

Another significant research area related to learning with data of different quality, is that of Multi-Fidelity models (MFMs)(Fernández-Godino, 2023; Peherstorfer et al., 2018). MFMs are models designed to learn from several sets of training data having different levels of quality. These approaches are typically used to learn from different granularity of numerical simulations and real measurements in physical experiments (Christen & Fox, 2005; Meng et al., 2021; Tonolini et al., 2020), although more common learning settings, such as classification have also been explored (Costabal et al., 2019; Chen et al., 2022). MFMs often use uncertainty quantification in order to capture model confidence, in particular for the higher fidelity data, which is often sparse (Meng et al., 2021; Peherstorfer et al., 2018). We draw inspiration from this key feature of MFMs to build our fine-tuning strategy, which learns from sparse clean data with a deep ensemble and capture model uncertainty.

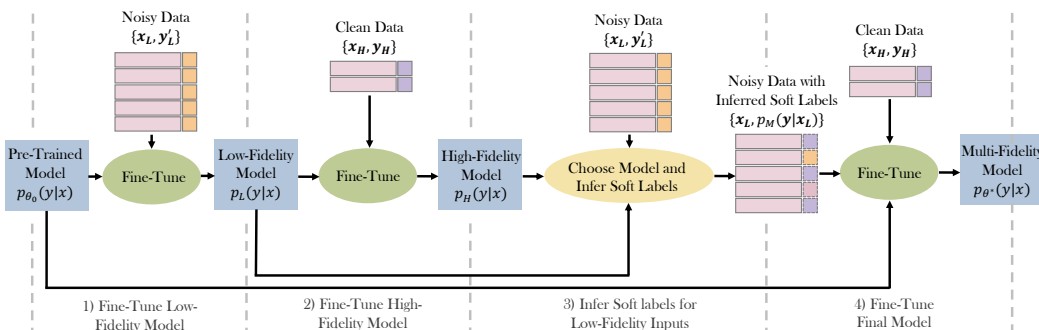

Figure 2: Multi-Fidelity Fine-Tuning in four stages. **1)** Data with noisy labels $\mathcal{D}_L = \{\mathbf{x_L}, \mathbf{y'_L}\}$ is used to fine-tune a pre-trained language model $p_{\theta_0}(y|x)$, resulting in a Low-Fidelity classifier $p_L(y|x)$. **2)** This low-fidelity model is fine-tuned further with available clean data $\mathcal{D}_H = \{\mathbf{x_H}, \mathbf{y_H}\}$ to obtain the high-fidelity classifier $p_H(y|x)$. **3)** For each input in the low-fidelity set $x_{L,i} \in \mathbf{x_L}$, the model expected to give the highest log likelihood between $p_L(y|x)$ and $p_H(y|x)$ is used to infer class probabilities, obtaining the set of soft labels $p_M(y|x_{L,i})$. **4)** The final multi-fidelity model $p_{\theta^*}(y|x)$ is fine-tuned from the pre-trained model $p_{\theta_0}(y|x)$ using all inputs $[\mathbf{x_L}, \mathbf{x_H}]$ and combined soft labels and high-fidelity hard labels as targets $[p_M(\mathbf{y}|\mathbf{x_L}), \mathbf{y_H}]$.

## 2.3 Fine-Tuning Language Models with Mixed Quality Data

Fine-tuning pre-trained language models with data of mixed quality is ubiquitous to many settings, including augmenting clean data with weakly supervised data (Li et al., 2021a; Lu & Radha, 2023; Yu et al., 2020b), active learning in noisy labels settings (Zhang et al., 2024; Goel et al., 2022) and fine-tuning with both human and LLM labeled or generated data (Wang et al., 2023a; Zhang et al., 2024; Meng et al., 2023). In some settings, clean and noisy data are simply aggregated in the fine-tuning set (Zhang et al., 2024). Other methods aggregate the two sets, but assigning a different cost weight to clean data (Wang et al., 2023a), while some approaches learn the noise process in different ways and incorporate it when training on noisy labels Jindal et al. (2019); Zhuang et al. (2023). Kim et al. (2024) have recently proposed to exploit the robustness properties of parameter efficient fine-tuning to learn from mixed quality data. Using LLMs to assist the noise cleaning process has also been explored Wang et al. (2023b). An effective and relatively simple approach, similar to task adaptive pre-training (Gururangan et al., 2020; Shi et al., 2023), domain adaptation (Chronopoulou et al., 2019) and transfer learning (Chronopoulou et al., 2019; Hedderich et al., 2020), first fine-tunes with the noisy data, and subsequently continues fine-tuning with the clean data (Li et al., 2021b; Tamkin et al., 2020; Zhu et al., 2022; Li et al., 2022). We also adopt this strategy as part of our approach, however, preventing over-fitting to clean data with our multi-fidelity strategy.

# 3 Multifidelity Fine-tuning (MFFT)

## 3.1 Problem Description

We consider the problem of fine-tuning a pre-trained language model for classification with two sets of task specific training data; a low-fidelity set $\mathcal{D}_L$ of $N$ examples $x_{L,i} \in \mathbf{x_L}$ with noisy labels $y'_{L,i} \in \mathbf{y'_L}$ and a high-fidelity set $\mathcal{D}_H$ of $M$ examples $x_{H,i} \in \mathbf{x_H}$ with clean labels $y_{H,i} \in \mathbf{y_H}$, where typically $M << N$. We assume the noisy labels $\mathbf{y'_L}$ to be the result of a noise process $y'_{L,i} = \eta(x_{L,i}, y_{L,i})$ which depends on both inputs $x_{L,i}$ and hidden ground-truth labels $y_{L,i}$. Given the two sets $\mathcal{D}_L$ and $\mathcal{D}_H$, we aim to fine-tune a language model with pre-trained weights $\theta_0$ to obtain a language classifier $p_{\theta^*}(y|x)$ which can perform the task of interest. Formally, our objective is to maximize the log likelihood assigned by the model $p_\theta(y|x)$ to clean data from the target distribution $p(x)p(y|x)$:

$$\theta^* = \arg\max_\theta \mathbb{E}_{p(x)p(y|x)} \log p_\theta(y|x). \tag{1}$$

Here $p(x)$ is the expected distribution of inputs at test time, which we assume our training inputs $\mathbf{x_H}$ and $\mathbf{x_L}$ to belong to, and $p(y|x)$ is the true input-labels mapping we aim to capture, which we do not

have access to. The problem we address is how to optimally exploit the noisy and clean sets $\mathcal{D}_L$ and $\mathcal{D}_H$ to fine-tune the language model $p_\theta(y|x)$ from pre-trained and approximately maximize Eq. 1.

## 3.2 OVERVIEW

We start by fine-tuning the pre-trained model $p_{\theta_0}(y|x)$ in two stages. First, we fine-tune the model on low-fidelity data $\mathcal{D}_L$. Second, we continue fine-tuning with high-fidelity data $\mathcal{D}_H$ to obtain a high-fidelity model $p_H(y|x)$. This sequential fine-tuning approach is common in transfer learning and semi-supervised learning with pre-trained language models (Yu et al., 2020a; Qin et al., 2022; Gururangan et al., 2020; Zhu et al., 2022; Shi et al., 2023). We find that it is as effective in our multi-fidelity scenario, providing a competitive baseline itself. However, with scarce high-fidelity data $\mathcal{D}_H$, $p_H(y|x)$ still over-fits, losing information gained during the first fine-tuning stage; a phenomenon sometimes referred to as catastrophic forgetting (Chen et al., 2020; Kotha et al., 2023). This results in $p_H(y|x)$ to be accurate for some inputs $x$ that are somewhat similar to high-fidelity examples $\mathbf{x_H}$, but inaccurate, and especially poorly calibrated, for others.

To address the aforementioned problem, we introduce a second version of the model by freezing the weights after the first fine-tuning stage. The resulting model is fine-tuned on low-fidelity data $\mathcal{D}_L$ only and we name it accordingly the low-fidelity model $p_L(y|x)$. This model does not over-fit, as it is fine-tuned on abundant examples, but its accuracy is limited by the data noise in $\mathcal{D}_L$. Our strategy is to use $p_L(y|x)$ as a fall-back for $p_H(y|x)$ for those inputs where the latter is expected to be less accurate than the former. Because we do not want to load and run both models at inference time, we also distill the resulting system into a single final fine-tuned language classifier $p_{\theta^*}(y|x)$. Our MFFT approach is summarized in four stages, schematically illustrated in Figure 2:

1. Fine-tune the pre-trained model $p_{\theta_0}(y|x)$ with low-fidelity data $\mathcal{D}_L$ to obtain $p_L(y|x)$.

2. Continue fine-tuning $p_L(y|x)$ with high-fidelity data $\mathcal{D}_H$ to obtain $p_H(y|x)$.

3. For each input in the low-fidelity set $x_{L,i} \in \mathbf{x_L}$, estimate the expected log likelihoods of both $p_H(y|x_{L,i})$ and $p_L(y|x_{L,i})$ and infer soft labels $p_M(y|x_{L,i})$, using the model expected to perform best.

4. Fine-tune the pre-trained model $p_{\theta_0}(y|x)$ with all inputs $[\mathbf{x_L}, \mathbf{x_H}]$ and combined inferred soft labels and high-fidelity hard labels as targets $[p_M(\mathbf{y}|\mathbf{x}_L), \mathbf{y_H}]$, obtaining the final model $p_{\theta^*}(y|x)$.

## 3.3 LOW- AND HIGH-FIDELITY MODEL FINE-TUNING

A key consideration in fine-tuning high and low fidelity models for our strategy is that we need to estimate their accuracy for new inputs to choose which one to use. As $p_L(y|x)$ is trained with abundant noisy data $\mathcal{D}_L$, its accuracy predominantly depends on data noise and, while it is not possible to estimate this from model output alone, we can approximately estimate it using clean examples in $\mathcal{D}_H$ as evaluation points (Hendrycks et al., 2018; Patrini et al., 2017). Contrarily, the accuracy of $p_H(y|x)$ depends predominantly on model mis-specification, or lack of knowledge, as it was fine-tuned on clean, but scarce data $\mathcal{D}_H$. This means that, to detect for which inputs $x$ the high-fidelity model is expected to be inaccurate, we need to construct $p_H(y|x)$ to obtain accurate uncertainty estimation with respect to its reducible error, i.e., due to lack of data.

**Fine-Tune Ensemble:** To obtain models that give accurate uncertainty estimation, we adopt a deep ensemble approach. We repeat the two-stages fine-tuning with different random seeds, obtaining $K$ fine-tuned models $p(y|x, \theta_{L,k})$ and $K$ fine-tuned models $p(y|x, \theta_{H,k})$. At inference time, the high and low fidelity label probabilities are obtained by aggregating their outputs:

$$p_L(y|x) = \frac{1}{K}\sum_k^K p(y|x, \theta_{L,k}), \quad p_H(y|x) = \frac{1}{K}\sum_k^K p(y|x, \theta_{H,k}). \tag{2}$$

Deep ensembles offer an effective way to capture uncertainty with respect to reducible error (Lakshminarayanan et al., 2017; Abdar et al., 2021; Rahaman et al., 2021). However, obtaining effective deep ensembles from pre-trained models is challenging, as the starting weights are fixed and cannot be randomly initialized for each model, leading to poor diversification and inaccurate uncertainty

estimation (Mustafa et al., 2020; Matthews & Lillis, 2022). To address this problem in our multi-fidelity setting, we exploit the abundance of low-fidelity data. We split the low-fidelity data into $K$ subsets $\mathcal{D}_{L,k}$ and use each one separately to perform the first stage of sequential fine-tuning. In the second stage, all models are then fine tuned on the entire high fidelity set $\mathcal{D}_H$. We found this strategy to greatly improve diversification of the ensembles, and hence model uncertainty quantification of $p_H(y|x)$, while not significantly impacting individual predictive performance of either low or high fidelity models.

### 3.4 SOFT LABEL INFERENCE

Having trained the low-fidelity and high-fidelity models $p_L(y|x)$ and $p_H(y|x)$, we need to choose which one to use during inference given a new input $x$. This requires to estimate which model will maximize our objective of Equation 1, given a new input $x$:

$$\alpha^* = \arg\max_\alpha \mathbb{E}_{p(y|x)} \log p_H(y|x)^\alpha p_L(y|x)^{1-\alpha}, \quad \alpha \in [0,1]. \tag{3}$$

Here $\alpha$ is a binary parameter, modeling the choice of either high or low fidelity model. Making this choice comes down to estimating the expected log likelihood for each model and selecting the one with the highest result.

**Estimate Log Likelihood of High-Fidelity Model:** Assuming the inference by each individual model $p(y|x, \theta_{H,k})$ in the high-fidelity ensemble is an un-biased estimator of the true mapping $p(y|x)$, we can estimate the log likelihood of the high fidelity model as:

$$\mathbb{E}_{p(y|x)} \log p_H(y|x) \approx -\mathcal{H}[p_H(y|x)], \tag{4}$$

Here $\mathcal{H}[p_H(y|x)]$ is the entropy of the high-fidelity model. A full derivation is given in Appendix A.1. This means that, if $p_H(y|x)$ is well calibrated with respect to its model error, we can use its entropy to infer expected log likelihood for a new input $x$.

**Estimate Log Likelihood of Low-Fidelity Model:** Unlike for the high-fidelity ensemble, we cannot assume that models $p(y|x, \theta_{L,k})$ trained on the low fidelity data $\mathcal{D}_L$ are unbiased estimators of the true mapping $p(y|x)$. This is because the training labels $\mathbf{y'_L}$ are affected by an unknown noise process, e.g., LLM hallucinations, which may be bias. As a result, the entropy of $p_L(y|x)$ is not expected to be a good estimator for the log likelihood of the low fidelity model. However, we can use the clean data points in $\mathcal{D}_H$ to estimate the average expected log likelihood over inputs $x$:

$$\mathbb{E}_{p(y|x)} \log p_H(y|x) \approx \frac{1}{M} \sum_i^M \log p_L(y_{H,i}|x_{H,i}) = \log p_L(y_H|x_H). \tag{5}$$

A full derivation is presented in Appendix A.2. This estimate approximates the log likelihood of the low-fidelity model as the marginal over all inputs. Some works make more granular estimations of this likelihood, e.g., by learning class probabilities mappings for each class (Hendrycks et al., 2018). However, in the low clean data regimes we target (tens of examples in $\mathcal{D}_H$), we choose to make this coarser, but more robust approximation. The resulting average likelihood can then be used as a threshold on the entropy of the high-fidelity model and choose whether to use the high or low fidelity model for inference.

**Compare Expected Log Likelihoods and Choose a Model:** With the estimates of high and low fidelity log likelihoods detailed above, we can choose which model to use to predict soft labels $p_M(\mathbf{y}|\mathbf{x}_L)$ over all the low fidelity inputs $\mathbf{x}_L$:

$$p_M(y|x_{L,i}) = p_H(y|x_{L,i})^{\alpha^*} p_L(y|x_{L,i})^{1-\alpha^*} \approx \begin{cases} p_H(y|x_{L,i}), & \text{if } -\mathcal{H}[p_H(y|x)] \geq \log p_L(y_H|x_H) \\ p_L(y|x_{L,i}), & \text{if } -\mathcal{H}[p_H(y|x)] < \log p_L(y_H|x_H). \end{cases} \tag{6}$$

A detailed derivation is provided in Appendix A.3. The choice of model is made by evaluating the entropy of the high-fidelity model $\mathcal{H}[p_H(y|x)]$ and using the constant $\log p_L(y_H|x_H)$ as a threshold.

### 3.5 FINAL MODEL FINE-TUNING

To obtain the final multi-fidelity model, we fine-tune the pre-trained model $p_{\theta_0}(y|x)$ using all available inputs aggregated $[\mathbf{x_L}, \mathbf{x_H}]$. As corresponding training labels, we use the available high-fidelity

labels $\mathbf{y_H}$ as targets for high-fidelity inputs $\mathbf{x_H}$ and the multi-fidelity soft labels $p_M(\mathbf{y}|\mathbf{x_L})$ as targets for the low-fidelity inputs $\mathbf{x_L}$. In particular, we maximize the following cross-entropy objective:

$$\theta^* = \arg\max_{\theta} \frac{1}{N+M} \Big[ \sum_i^M \log p_\theta(y_{H,i}|x_{H,i}) + \sum_i^N \sum_j^C p_M(y_j|x_{L,i}) \log p_\theta(y_j|x_{H,i}) \Big]. \quad (7)$$

Here $C$ is the number of classes for the task. The resulting fine-tuned model $p_{\theta^*}(y|x)$ is identical in structure to the pre-trained model $p_{\theta_0}(y|x)$ and can be deployed by simply loading the fine-tuned weights $\theta^*$.

## 4 EXPERIMENTS

We evaluate MFFT across several data sets, models and against different baselines. We perform experiments in two different label noise scenarios:

- **Simulated Noise:** In Section 4.2, we fine-tune models with clean data, together with artificially corrupted data. We find that MFFT gives competitive or better negative log likelihood (NLL) in almost all experiments and, on average, outperforms the best baseline by $4.6\%$ in F1, while reducing expected calibration error (ECE) by $64\%$.

- **LLM Generated:** In Section 4.3, we fine-tune with clean data, together with LLM generated data. MFFT obtains competitive or better NLL in all experiments and, on average, outperforms the best baselines by $1.8\%$ in F1, while reducing ECE by $40\%$.

### 4.1 EXPERIMENTAL SETTINGS AND BASELINES

We test MFFT using six benchmark text classification data sets; AGNews (Zhang et al., 2015b), DBPedia 14 (Zhang et al., 2015a), GLUE-SST2, GLUE-QQP (Wang et al., 2019), TREC (Li & Roth, 2002) and Yahoo Answers (Adamic et al., 2008). More details about these data sets and tasks are given in Appendix B.1. In all our experiments, we take $5,000$ examples to construct the training set and $1,000$ examples for testing. The training set is split into two sub-sets; a high-fidelity set $\mathcal{D}_H$, for which the labels are directly extracted from the original source data, and a low-fidelity set $\mathcal{D}_L$, for which the labels are either artificially corrupted, in the simulated noise experiments, or inferred using an LLM, in the LLM experiments. In all experiments, we compare MFFT to six baseline approaches used in similar settings in related work:

- **High:** We discard the low-fidelity data $\mathcal{D}_L$ and fine-tune the pre-trained language model solely with high-fidelity data $\mathcal{D}_H$.

- **Together:** High and Low fidelity data sets are simply aggregated into a single data set which is used to fine-tune the pre-trained language model. This approach has been used for the fine-tuning component of several works involving mixed quality data (Zhang et al., 2024; Li et al., 2021a; Lu & Radha, 2023).

- **Cost Adjustment:** Similarly to the above, the model is trained on all available data together. However, a higher weight is assigned to the cost from high-fidelity data following Wang et al. (2023a).

- **Low-High:** A domain adaptation approach, where the model is initially fine-tuned on low-fidelity data $\mathcal{D}_L$ and then on $\mathcal{D}_H$. This approach is adopted in several multi-fidelity learning works (Aydin et al., 2019; Li et al., 2021b; 2022) and can be considered a special case of task adaptive pre-training (TAPT) (Gururangan et al., 2020; Shi et al., 2023).

- **High-Low:** The language model is first fine-tuned on high-fidelity data $\mathcal{D}_H$ and then on low-fidelity data $\mathcal{D}_L$, with label smoothing and temporal ensembling (Meng et al., 2023).

- **Noise Correction:** The high-fidelity points are used to learn a linear noise process from clean to noisy labels. This mapping is applied to the cost function for low-fidelity examples, which are then used together with high-fidelity ones to fine-tune the final model (Hendrycks et al., 2018; Jindal et al., 2019).

More details about the implementation of these baselines are given in appendix B.2. We test MFFT and all baselines with three pre-trained language models: BERT (Vaswani et al., 2017), RoBERTa

|  | High | Together | Cost Adj. | Low-High | High-Low | Noise Cor. | MFFT |
|---|---|---|---|---|---|---|---|
| **BERT** | | | | | | | |
| AGNews | 0.990 | 0.778 | 0.742 | 0.901 | 0.833 | 0.944 | **0.357** |
| DBPedia 14 | 0.437 | 0.569 | 0.535 | **0.097** | 0.808 | 0.913 | **0.079** |
| GLUE-SST2 | 1.666 | 0.390 | 0.457 | 0.921 | 0.470 | 0.450 | **0.303** |
| GLUE-QQP | 2.736 | 0.928 | 0.848 | 2.413 | **0.730** | 0.667 | 0.724 |
| TREC | 1.299 | 0.731 | 0.678 | 0.517 | 0.838 | 0.990 | **0.305** |
| Yahoo | 3.238 | 1.378 | 1.379 | 2.125 | 1.453 | 1.736 | **1.343** |
| **RoBERTa** | | | | | | | |
| AGNews | 1.091 | 0.937 | 0.966 | 1.092 | 0.945 | 0.936 | **0.367** |
| DBPedia 14 | 0.730 | 0.513 | 0.489 | **0.118** | 0.676 | 0.723 | **0.091** |
| GLUE-SST2 | 1.616 | 0.830 | 0.736 | 1.883 | 0.634 | 0.623 | **0.364** |
| GLUE-QQP | 2.851 | 0.668 | **0.658** | 2.887 | **0.651** | 0.693 | **0.611** |
| TREC | 1.963 | 0.576 | 0.593 | 0.496 | 0.760 | 0.775 | **0.238** |
| Yahoo | 3.246 | 1.506 | 1.533 | 2.669 | 1.608 | 1.857 | **1.341** |
| **GPT-2-Medium** | | | | | | | |
| AGNews | 1.300 | 0.741 | 0.741 | 1.553 | 0.807 | 0.856 | **0.391** |
| DBPedia 14 | 2.051 | 1.178 | 0.835 | 1.085 | 1.044 | 1.132 | **0.256** |
| GLUE-SST2 | 2.465 | 1.142 | 0.831 | 3.007 | 0.728 | **0.686** | **0.670** |
| GLUE-QQP | 2.826 | 0.756 | **0.714** | 3.756 | **0.701** | 0.694 | 0.931 |
| TREC | 2.309 | 1.343 | 1.387 | 3.288 | 1.361 | 1.453 | **0.705** |
| Yahoo | 4.560 | 1.938 | 1.718 | 4.595 | 1.706 | 1.942 | **1.479** |

Table 1: Negative log-likelihood (NLL) of different fine-tuning strategies with simulated label noise. Numbers in bold indicate best performance, or within statistical significance (p-value<0.05) of best performance across fine-tuning methods.

|  | Together | Cost Adj. | Low-High | High-Low | Noise Cor. | MFFT |
|---|---|---|---|---|---|---|
| NLL | 86.1% | 88.9% | 63.9% | 86.1% | 83.3% | **100.0%** |
| F1-Score | 19.5% | 27.8% | 75.0% | 55.6% | 52.8% | **100.0%** |
| ECE | 69.4% | 63.9% | 77.8% | 55.6% | 52.8% | **100.0%** |

Table 2: Percentage of experiments with simulated label noise (36 total) in which each fine-tuning approach resulted in competitive or better performance compared to fine-tuning solely with the small set of trusted data.

(Liu et al., 2019) and GPT-2-Medium (Radford et al., 2019). All experiments are repeated five times for statistical significance.

## 4.2 SIMULATED NOISE

With each data set, we simulate noise in the training set by artificially corrupting labels. We follow the simulation approach of Hendrycks et al. (2018) to generate noise through the specification of a noise process (details in Appendix B.3). We report here results for simulation settings such that labels in the noisy set have 0.3 probability of being incorrect. Results for different settings and ablations over noise process parameters are reported in Appendix C. We keep 50 training examples with the original clean labels to form the high-fidelity set $\mathcal{D}_H$. We then fine-tune models with all baselines and MFFT. We measure negative log-likelihood as the main metric, as it captures both classification performance and calibration and is the target in our formulation (Equation 1). Results are shown in Table 1. We also record macro F1-score and expected calibration error (ECE) as independent measures of classification performance and calibration. These are shown for three datasets in Figure 3. We report more experimental results with higher noise in Appendix C.2), as well as ablations over different experimental conditions in Appendix C.3.

In Table 1, we observe that models fine-tuned with MFFT resulted in better or competitive NLL in all but one experiment (GLUE-QQP with GPT-2-Medium) and statistically better than any other in 66% of cases. The overall most competitive baseline is Low-High (transfer learning approach),

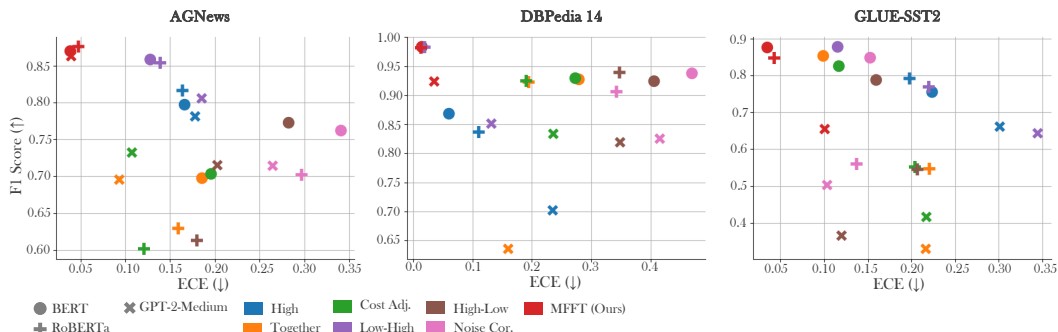

Figure 3: Macro F1-Score vs. expected calibration error (ECE). Favorable performance is in the top left corner of the graphs, where classification performance is high (high F1-score) and, simultaneously, calibration error is low (low ECE).

which MFFT outperforms in both classification performance, with an average improvement in F1 of $0.034$ ($+4.6\%$), and especially calibration, with an average reduction in ECE of $0.146$ ($-64\%$). We observed similar results with a higher level of simulated noise (Appendix C.2). We also note in Table 2 that MFFT is competitive or better than fine-tuning with high-fidelity data only (High) across all experiments and all considered metrics (36 total experiments). This means that, using MFFT, fine-tuned models always benefit from additional data, despite their lower quality. This is not true for any of the baselines.

## 4.3 REAL NOISE FROM LLM GENERATED DATA

We consider the setting where we have access to a small labeled data set $\mathcal{D}_H$ and use a pre-trained large language model (LLM) to generate more data to fine-tune a classifier. With the three data sets AGNews, GLUE-SST2 and TREC, we explore two common types of LLM data generation: i) labeling through prompting, where we assume access to an unsupervised data set $\mathbf{x_L}$ and infer labels $\mathbf{y'_L}$ with a pre-trained LLM, using examples from $\mathcal{D}_H$ for in-context learning (Thapa et al., 2023; Ding et al., 2024) (details in Appendix B.4). ii) Data augmentation, where we only have access to the small data set $\mathcal{D}_H$ and use the LLM to generate new inputs and outputs, using $\mathcal{D}_H$ as instruction examples (Ding et al., 2024; Meng et al., 2023; Wang et al., 2023a) (details in Appendix B.5). We use 100 labeled examples as the data set $\mathcal{D}_H$ and generate $5,000$ more with the LLM with either method. We use both Mistral-7B-Instruct-v0.3 (Jiang et al., 2023) and Gemma-7b-it (Team et al., 2024) as pre-trained LLMs. With both original data $\mathcal{D}_H$ and generated data $\mathcal{D}_L$, we fine-tune RoBERTa using MFFT and all baselines. We report NLL in table 3 and F1 vs. ECE in Figure 4.

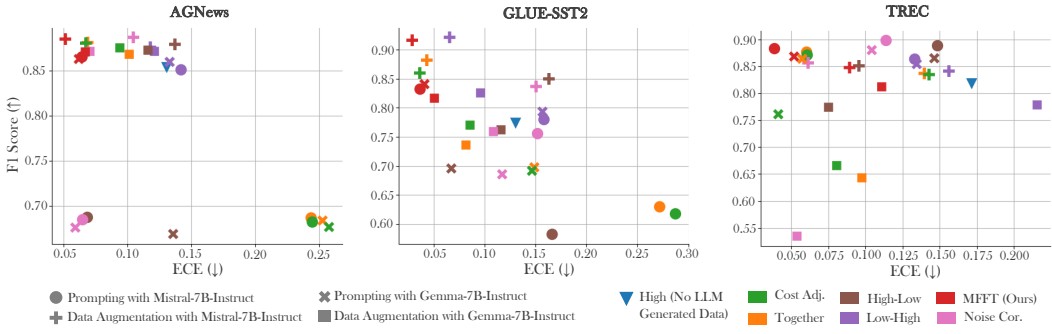

Figure 4: Macro F1-Score vs. expected calibration error (ECE) for RoBERTa fine-tuned on 100 clean examples and $5,000$ examples generated with LLMs. Favorable performance is in the top left corner of the graphs, where classification performance is high (high F1-score) and, simultaneously, calibration error is low (low ECE).

Table 3 shows the results for models fine-tuned with MFFT resulted in better or competitive NLL in all experiments. In Figure 4, MFFT performs consistently well, with relatively high F1 and low ECE,

| Data Set | High | Together | Cost Adj. | Low-High | High-Low | Noise Cor. | MFFT |
|---|---|---|---|---|---|---|---|
| **Prompting with Mistral-7B-Instruct** | | | | | | | |
| AGNews | 0.923 | 1.334 | 1.241 | 1.123 | 0.844 | 0.616 | **0.409** |
| GLUE-SST2 | 1.406 | 0.373 | 0.391 | 1.097 | 0.381 | 0.339 | **0.294** |
| TREC | 0.917 | 1.189 | 1.146 | 1.155 | 1.226 | 0.755 | **0.338** |
| **Data Augmentation with Mistral-7B-Instruct** | | | | | | | |
| AGNews | 0.948 | **0.406** | **0.390** | 0.915 | 0.463 | 0.416 | **0.359** |
| GLUE-SST2 | 1.455 | 0.718 | 0.728 | 1.297 | **0.412** | **0.401** | 0.451 |
| TREC | 0.927 | 0.395 | 0.441 | 0.484 | 0.514 | 0.472 | **0.228** |
| **Prompting with Gemma-7B-Instruct** | | | | | | | |
| AGNews | 0.933 | 1.386 | 1.355 | 1.022 | 0.913 | 0.663 | **0.426** |
| GLUE-SST2 | 1.413 | 0.399 | 0.428 | 0.999 | 0.414 | **0.364** | **0.336** |
| TREC | 0.906 | 1.083 | 1.008 | 1.088 | 1.002 | 0.894 | **0.341** |
| **Data Augmentation with Gemma-7B-Instruct** | | | | | | | |
| AGNews | 0.919 | 0.558 | 0.514 | 0.954 | **0.466** | **0.475** | **0.414** |
| GLUE-SST2 | 1.426 | 0.602 | **0.567** | 1.817 | **0.518** | 0.624 | **0.565** |
| TREC | 0.988 | 0.486 | 0.482 | 0.701 | 0.525 | 0.558 | **0.340** |

Table 3: Negative log-likelihood (NLL) of different fine-tuning strategies to combine a large amount (5,000 examples) of LLM labeled or generated data and a small amount (100 examples) of trusted clean data. Numbers in bold indicate best performance, or within statistical significance of best performance across fine-tuning methods.

| | Together | Cost Adj. | Low-High | High-Low | Noise Cor. | MFFT |
|---|---|---|---|---|---|---|
| NLL | 75.0% | 75.0% | 58.3% | 91.7% | 95.8% | **100.0%** |
| F1-Score | 62.5% | 62.5% | 83.3% | 58.3% | 62.5% | **100.0%** |
| ECE | 70.8% | 75.0% | 66.7% | 87.5% | 87.5% | **100.0%** |

Table 4: Percentage of experiments with LLM generated data (24 total) in which each fine-tuning approach resulted in competitive or better performance compared to fine-tuning solely with the small set of trusted data.

across different LLMs, tasks and training data generation modalities. Conversely, other baselines often under-perform in either F1 or ECE. The most competing baseline in classification performance (average F1-score) is Low-High, which MFFT outperforms by $0.016$ in F1-score ($+1.84\%$) and $0.078$ in ECE ($-57.5\%$). The most competing baseline in calibration (average ECE) is Noise Correction, which MFFT outperforms by $0.081$ in F1-score ($+10.5\%$) and $0.039$ in ECE ($-40.4\%$). We also conduct experiments with a smaller clean data set $\mathcal{D}_H$ of 50 examples, in which we observe analogous results, shown in Appendix C.4. As for the simulated noise experiments, we observe in Table 4 that MFFT always matches or outperforms using trusted data only (24 total experiments), while this is not the case for any baseline. This means that, with MFFT, we can always benefit from LLM generated data when fine-tuning the end model.

## 5 CONCLUSION

We proposed Multi-Fidelity Fine-Tuning (MFFT), a novel method to fine-tune pre-trained language models with a small amount of clean trusted data (high-fidelity) and a larger amount of noisy data (low-fidelity). MFFT exploits knowledge derived from noisy data and knowledge derived from trusted data differently, implicitly learning when the latter can be used to infer and when, conversely, predicting from the former is expected to be more accurate. This leads to models fine-tuned with MFFT to consistently benefit from additional noisy data, while other methods carry the risk of degrading performance compared to using trusted data only. In our experiments, MFFT consistently outperformed all baselines across experiments, especially in calibration, with macro F1-score improvements of $2-4\%$ and expected calibration error (ECE) reductions of $40-60\%$ compared to the best baselines.

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

## A PROOFS AND DERIVATIONS

### A.1 DERIVATION OF EXPECTED HIGH-FIDELITY LOG LIKELIHOOD AS MODEL ENTROPY

Assuming the inference by each model $p_H(y|x, \theta_k)$ in the high-fidelity ensemble are un-biased estimators of the true mapping $p(y|x)$:

$$
\begin{aligned}
\mathbb{E}_{p(y|x)} \log p_H(y|x) &\approx \frac{1}{K} \sum_k^K \mathbb{E}_{p(y|x,\theta_k)} \log p_H(y|x) \\
&= \frac{1}{K} \sum_k^K \int p(y|x, \theta_k) \log p_H(y|x) dy \\
&= \int \frac{1}{K} \sum_k^K p(y|x, \theta_k) \log p_H(y|x) dy \\
&= \int p_H(y|x) \log p_H(y|x) dy \\
&= -\mathcal{H}(p_H(y|x))
\end{aligned}
\tag{8}
$$

### A.2 DERIVATION OF EXPECTED LOW-FIDELITY LOG LIKELIHOOD

We coarsely approximate the log likelihood of the low fidelity model $p_L(y|x)$ as its marginal over all inputs $x$, i.e., we expect the log likelihood of the low-fidelity model to be approximately constant with respect to $x$:

$$
\mathbb{E}_{p(y|x)} \log p_L(y|x) \approx \mathbb{E}_{p(x)} \mathbb{E}_{p(y|x)} \log p_L(y|x)
\tag{9}
$$

Now we can use the available high-fidelity samples $x_{H,i} \in \mathbf{x_H}$ and $y_{H,i} \in \mathbf{y_H}$ as samples from the true distributions $x_{H,i}, y_{H,i} \sim p(x)p(y|x)$:

$$
\begin{aligned}
\mathbb{E}_{p(x)} \mathbb{E}_{p(y|x)} \log p_L(y|x) &\approx \mathbb{E}_{x \in \mathbf{x_H}} \mathbb{E}_{y \in \mathbf{y_H}} \log p_L(y|x) \\
&= \frac{1}{M} \sum_i^M \log p_L(y_{H,i}|x_{H,i})
\end{aligned}
\tag{10}
$$

### A.3 DERIVATION OF MODEL CHOICE

Our objective is to approximately maximize the expected log likelihood of equation 3:

$$
\begin{aligned}
&\mathbb{E}_{p(y|x)} \log p_H(y|x)^\alpha p_L(y|x)^{1-\alpha} \\
&= \alpha \mathbb{E}_{p(y|x)} \log p_H(y|x) + (1-\alpha) \mathbb{E}_{p(y|x)} \log p_L(y|x) \\
&\approx -\alpha \mathcal{H}[p_H(y|x)] + (1-\alpha) \log p_L(y_H|x_H) \quad \text{from eq. 4 and 5}
\end{aligned}
\tag{11}
$$

As the two terms are linearly added, maximizing equation 11 for the parameter $\alpha \in [0, 1]$ results in choosing either $\alpha = 0$ or $\alpha = 1$, depending on which term is greater:

$$
\alpha^* = \begin{cases} 1, & \text{if } -\mathcal{H}[p_H(y|x)] \geq \log p_L(y_H|x_H) \\ 0, & \text{if } -\mathcal{H}[p_H(y|x)] < \log p_L(y_H|x_H). \end{cases}
\tag{12}
$$

Applying this choice to the combined soft labels $p_M(\mathbf{y}|\mathbf{x}_L)$ we obtain:

$$
p_M(y|x_{L,i}) = p_H(y|x_{L,i})^{\alpha^*} p_L(y|x_{L,i})^{1-\alpha^*} \approx \begin{cases} p_H(y|x_{L,i}), & \text{if } -\mathcal{H}[p_H(y|x)] \geq \log p_L(y_H|x_H) \\ p_L(y|x_{L,i}), & \text{if } -\mathcal{H}[p_H(y|x)] < \log p_L(y_H|x_H). \end{cases}
\tag{13}
$$

# B  ADDITIONAL EXPERIMENTAL DETAILS

## B.1  DATASETS DETAILS

We evaluate MFFT and all baselines using 6 NLP tasks, spanning various number of classes and types of tasks. These are:

- **AGNews:** Topic modeling of news extracts. The data set consists of extracts from news passages to be classified into one of four classes: "World", "Sports", "Business" and "Science and Technology".

- **DBPedia 14:** Topic modeling of extracts from Wikipedia articles to be classified into one of 14 classes: "Company", "Educational Institution", "Artist", "Athlete", "Office Holder", "Mean of Transport", "Building", "Natural Place", "Village", "Animal", "Plant", "Album", "Film" and "Written Work".

- **GLUE-SST2:** Sentiment analysis of extracts from movie reviews to be classified as either negative or positive.

- **GLUE-QQP:** Data set of pairs of questions to be classified as either duplicates of each other or not duplicates.

- **TREC:** Topic modeling of questions. We use the coarse labels of the data set for our evaluation. The data set contains questions to be classified into one of 6 classes: "Abbreviation", "Entity", "Description", "Human Being", "Location", "Numeric Value".

- **Yahoo Answers:** Topic modeling of questions from Yahoo Answers. Data set contains questions to be classified into one of 10 classes: "society", "science", "health", "education", "computers", "sports", "business", "entertainment", "family" and "politics".

## B.2  BASELINES AND MFFT IMPLEMENTATION DETAILS

We use a small validation set with original clean labels to perform early-stopping when fine-tuning models. As we consider scenarios where clean labels are scarce, we set the number of validation examples to 50. All fine-tuning steps are performed using the AdamW optimizer and an initial learning rate of $5 \times 10^{-5}$. The macro f1 score is computed with the validation set every 100 iterations and the model corresponding to the highest score is chosen (early stopping). All models are optimised using the standard cross-entropy cost with hard labels, and using equation 7 with soft or mixed labels. All validations of hyper-parameters were done using the TREC data set with simulated noisy labels and 50 clean training examples. Candidate values for the hyper-parameters were simply tried and the best performing on the validation set was chosen. Hyper-parameters were kept constant to these values optimized with the TREC data set for all experiments.

**High Only**

The pre-trained model is fine-tuned using only clean data $\mathcal{D}_H$ only. The model is fine-tuned with a cross-entropy cost for a fixed number of $1,000$ iterations.

**Together**

Clean and noisy data $\mathcal{D}_H$ and $\mathcal{D}_L$ are aggregated into a single data set. The pre-trained language model is then fine-tuned for one epoch on this data set.

**Cost Adjustment**

Analogously to the above, clean and noisy data is aggregated into a single data set and the model is fine-tuned for one epoch. However, the cross-entropy cost for clean data is weighted by 2.0. We defined this hyper-parameter by cross-validation as described above, validating performance for 1.5, 2.0, 3.0, 5.0 and 10.0.

**Low then High**

The pre-trained model is fine tuned for 1 epoch on the noisy data $\mathcal{D}_L$ and subsequently fine-tuned further for 100 iterations on the clean data $\mathcal{D}_H$.

**High then Low**

The pre-trained model is fine-tuned on clean data first $\mathcal{D}_H$ for 50 iterations (100 iterations resulted in lower performance, we believe because of catastrophic forgetting). Second, the model is fine tuned on noisy data $\mathcal{D}_L$, but introducing label smoothing and temporal averaging. Following Equation 5 in (Meng et al., 2023), we cross-validated $\epsilon = 0.1$, choosing from 0.01, 0.05, 0.1, and 0.2, $\lambda = 0.05$ choosing from 0.01, 0.05 and 0.1. Temporal averaging is done over 5 iterations.

**Noise Correction**

First the pre-trained model is fine-tuned with noisy data $\mathcal{D}_L$ for 1 epoch. Next, a linear mapping between true clean labels and inferences from this model is learned using the available clean data $\mathcal{D}_H$. This results into a matrix $C$ mapping vectors of clean labels probabilities to vectors of noisy labels probabilities. The pre-trained model is now trained using all data, but applying the matrix $C$ to outputs of the model before computing cross-entropy with noisy labels (see (Hendrycks et al., 2018) for details). This fine-tuning is performed for 2 epochs, as we found that fine-tuning to soft labels takes longer to converge.

**MFFT**

In all experiments, we construct MFFT with ensembles of 5 fine-tuned models. First we divide at random the clean data $\mathcal{D}_L$ into 5 equally sized subsets. As in the experiments we always use $5,000$ clean examples, this results into 5 sub-sets of $1,000$ examples each. We use each sub-set to fine-tune the model from pre-trained for 3 epochs and obtain the models $p(y|x, \theta_{L,k})$. Each model is then further fine-tuned with the entire clean set $\mathcal{D}_H$ for 100 iterations. Soft labels over the low-fidelity set $p_M(y|x_{L,i})$ are then computed as described in section 3.4. The final model is fine-tuned for 2 epochs from pre-trained using the cross-entropy cost of equation 7.

### B.3 Details of Noise Simulation

For our simulated experiments, we introduce noise by altering labels in the source data set with a stochastic process. In our simulation, we define two controllable parameters; noise level $l \in [0, 1]$ and noise bias $b \in [0, 1]$. Both of these are used in constructing a noise process matrix $M_\epsilon$ which maps clean labels to probabilities of noisy labels:

$$p(y'|y) = M_\epsilon p(y^*|y), \tag{14}$$

where $p(y'|y)$ is the probability of corrupted labels and $p(y^*|y)$ is the probability form of the ground-truth labels, i.e., a vector as long as the number of classes $N_c$, with one on the class corresponding to the ground-truth label $y$ and zeros everywhere else. The corrupted label $y'$ is then obtained by sampling from this distribution $y' \sim p(y'|y)$. The transition matrix $M_\epsilon$ is computed with the noise level parameter $l$ and the bias pareameter $b$ as follows:

$$M_\epsilon = (1-l)I(1+bR) + \frac{l}{N_c - 1}(\mathbf{1} - I)(1+bR). \tag{15}$$

Here $I$ is the $N_c \times N_c$ identity matrix, $\mathbf{1}$ is a $N_c \times N_c$ matrix of ones and $R$ is a $N_c \times N_c$ matrix where the elements are random uniform between zero and one. The matrix $M_\epsilon$ is first capped so that all elements are between zeor and one and secondly normalized so that the rows add up to one. With $b = 0$ (no bias), the matrix results in labels corrupted through equation 14 to be changed to another class with probability $l$ and staying the same with probability $1 - l$. The higher the value of $b$, the higher the randomness in the transition matrix, meaning that not all classes have equal probability to change and a given label $y$ has non-equal probability to change to each one of the other classes. This introduces label bias proportionally to $b$. In the simulated experiments of section 4.2, we fix $b = 0.3$ and test with two noise levels; $l = 0.3$ (results in section 4.2) and $l = 0.5$ (results in section C.2).

### B.4 Details of Experiments with LLM Prompting

As for the simulated data experiments, we take $5,000$ examples to construct the low-fidelity set $\mathcal{D}_L$. However, instead of artificially corrupting the labels provided with a noise process, we infer labels using the LLM with an in-context learning approach. We first construct a prompt to solve the given task, providing the input example, an instruction and a list of options. These prompts for each data-set are as follows:

**AGNews**

News Extract: <TEXT–INPUT>

Which one of the following topics does the above news extract fall under?
1) world
2) sports
3) business
4) science

<LABEL–OUTPUT>

**GLUE-SST2**

Question: <TEXT–INPUT>

What **is** the sentiment of the movie review extract above?
1) negative
2) positive

<LABEL–OUTPUT>

**TREC**

Text Extract: <TEXT–INPUT>

What **is** the text extract above about? Choose **from**:
1) entity
2) description
3) person
4) abbreviation
5) location
6) value

<LABEL–OUTPUT>

To form the in-context learning prompts, we use 5 labeled examples from the set small trusted set $\mathcal{D}_H$. We repeat experiments 5 times, each time re-drawing these 5 examples at random and keeping them fixed for inference over all unlabeled inputs $\mathbf{x_H}$. denoting the prompts detailed above for each data set as prompt(<TEXT-INPUT>, <LABEL-OUTPUT>), the in-context learning prompt is constructed as:

```
final_prompt = ''
for text_in , label in labelled_examples :
    final_prompt = final_prompt + prompt(text_in , label)
final_prompt = final_prompt + prompt(<TEST–INPUT>, '')
```

In this way, the LLM is prompted to provide an answer to the task instructions, using the input text <TEST-INPUT> as context and using 5 samples from the trusted labeled set as solved examples. To perform prompting, we compare the LLM next word output logits assigned to each of the candidate words for each class (numbered options in prompt format above) and pick the one with the highest. We repeat this operation over all inputs in $\mathbf{x_L}$ to obtain noisy labels $\mathbf{y'_l}$, together forming the low-fidelity set $\mathcal{D}_L$. We then fine-tune RoBERTa with $\mathcal{D}_L$ and $\mathcal{D}_H$ using MFFT and all baselines.

B.5    DETAILS OF EXPERIMENTS WITH LLM GENERATION

To generate examples and build the data set $\mathcal{D}_L$, we adopt an augmentation in-context learning approach, where, for each class, we provide a list of 5 examples of inputs taken from $\mathcal{D}_H$ in the prompt and let the LLM generate a new example to continue the list. The format is as follows:

<CLASS–SPECIFIC–AUGMENTATION–INSTRUCTION>
1) <EXAMPLE–1>
2) <EXAMPLE–2>

3) <EXAMPLE–3>
4) <EXAMPLE–4>
5) <EXAMPLE–5>
6)

Where the examples <EXAMPLE-i> are drawn at random from examples in $\mathbf{x_H}$ with the same associated label $y_h$ and <CLASS-SPECIFIC-AUGMENTATION-INSTRUCTION> varies depending on the data set and class. For each data set used and each class, these are as follows:

**AGNews**

'Make more extracts of world news examples like these ones'
'Make more extracts of sports news examples like these ones'
'Make more extracts of business news examples like these ones'
'Make more extracts of science news examples like these ones'

**GLUE-SST-2**

'Make more extracts of negative movie reviews like these ones'
'Make more extracts of positive movie reviews like these ones'

**TREC**

'Make more encyclopedia extracts examples about entities like these ones'
'Make more encyclopedia extracts examples about descriptions like these ones'
'Make more encyclopedia extracts examples about people like these ones'
'Make more encyclopedia extracts examples about abbreviations like these ones'
'Make more encyclopedia extracts examples about locations like these ones'
'Make more encyclopedia extracts examples about values like these ones'

The 5 examples provided in the prompt are re-drawn at random from $\mathbf{x_H}$ for every generation of a new example. We repeat this generation to obtain a fixed number of synthetic examples per class, such that the final data set $\mathcal{D}_L$ is balanced across classes and contains $5,000$ examples. We then fine-tune pre-trained RoBERTa on $\mathcal{D}_L$ and $\mathcal{D}_H$ with all baselines and MFFT.

# C  ADDITIONAL EXPERIMENTAL RESULTS

## C.1  SIMULATED NOISE F1 VS. ECE FOR ALL DATA SETS

We show in figure 5 the F1 vs. ECE plots of figure 3 for all data sets. The trends observed in figure 3 are observed across all tested data-sets, with models fine-tuned using MFFT appearing in the top-left corner of the plots and out-performing or matching baselines in most cases in both classification performance (F1-score) and calibration (ECE). In cases where a competing baseline shows marginally better performance in one of the two metrics, e.g., F1-score of Low-High on TREC, MFFT appreciably out-performs in the other, meaning that it maintains a favorable balance between classification performance and calibration.

## C.2  BENCHMARK EXPERIMENTS AT HIGHER NOISE LEVEL

We repeat the experiments of section 4.2 with a higher setting of noise level $l = 0.5$, i.e., on average, $50\%$ of labels are changed to a different class in the noise process. NLL results are shown in table 5 and F1-score vs. ECE plots are shown in figure 6.

Similarly to the results at moderate noise level of section 4.2, we observe appreciable improvement when fine-tuning models with MFFT compared to the baselines. Referring to table 5, MFFT was found to be the best or competitive in $72\%$ of experiments, while the best baseline according to NLL (noise correction) is competitive in only $22\%$ of cases. While for the moderate noise experiments of section 4.2 the best baseline was Low-High for all three metrics (NLL, F1 and ECE), with the higher noise, noise correction is the most competing baseline for NLL and ECE (better calibration), while Low-High remains the most competitive in terms of F1 score (classification performance). Compared

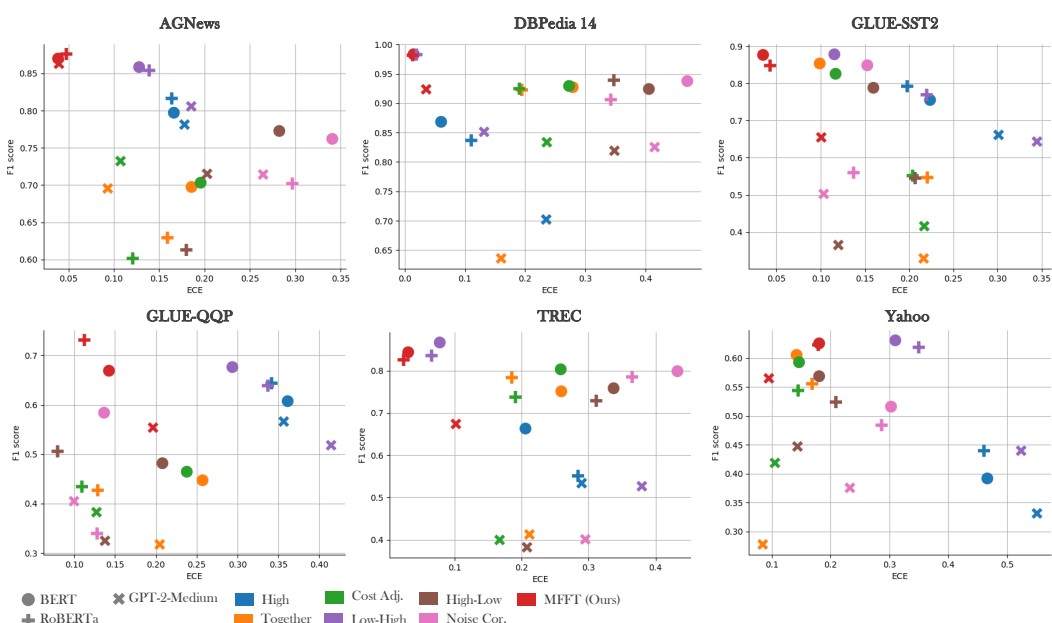

Figure 5: Macro F1-Score vs. expected calibration error (ECE) with moderate noise level $= 0.3$. Favorable performance is in the top left corner of the graphs, where classification performance is high (high F1-score) and, simultaneously, calibration error is low (low ECE).

| | **High** | **Together** | **Cost Adj.** | **Low-High** | **High-Low** | **Noise Cor.** | **MFFT** |
|---|---|---|---|---|---|---|---|
| **BERT** | | | | | | | |
| AGNews | 1.113 | 0.980 | 1.039 | 0.908 | 1.024 | 1.150 | **0.474** |
| DBPedia 14 | 0.596 | 2.018 | 1.952 | **0.359** | 1.296 | 2.145 | **0.385** |
| GLUE-SST2 | 2.440 | 0.701 | 0.711 | 2.054 | 0.668 | 0.657 | **0.597** |
| GLUE-QQP | 2.572 | 0.988 | 0.981 | 2.800 | 0.786 | **0.690** | 0.910 |
| TREC | 1.152 | 1.530 | 1.480 | 1.001 | 1.259 | 1.532 | **0.598** |
| Yahoo | 3.535 | **2.089** | **2.092** | 2.983 | 1.775 | 2.154 | 2.329 |
| **RoBERTa** | | | | | | | |
| AGNews | 1.313 | 0.797 | 0.759 | 0.952 | 0.894 | 0.972 | **0.667** |
| DBPedia 14 | 0.949 | 1.175 | 1.147 | 0.464 | 1.220 | 1.616 | **0.622** |
| GLUE-SST2 | 1.848 | 0.826 | 0.874 | 1.740 | 0.692 | 0.638 | **0.542** |
| GLUE-QQP | 2.704 | 1.309 | 1.375 | 2.650 | 0.837 | **0.688** | 1.008 |
| TREC | 1.728 | 1.169 | 1.245 | 1.261 | 1.180 | 1.413 | **0.650** |
| Yahoo | 3.520 | **1.823** | 1.872 | 3.186 | 1.742 | 2.109 | 2.228 |
| **GPT-2-Medium** | | | | | | | |
| AGNews | 1.491 | 1.171 | 1.126 | 2.340 | 1.121 | 1.190 | **0.523** |
| DBPedia 14 | 1.200 | 1.436 | 1.158 | 1.034 | 1.534 | 1.657 | **0.252** |
| GLUE-SST2 | 2.390 | 0.952 | 0.874 | 3.112 | **0.789** | 0.674 | 0.757 |
| GLUE-QQP | 3.104 | 0.892 | 0.912 | 3.449 | 0.802 | **0.690** | 0.949 |
| TREC | 4.430 | 1.611 | 1.644 | 4.275 | 1.507 | 1.579 | **0.947** |
| Yahoo | 5.028 | 2.221 | 2.145 | 5.351 | 2.249 | 2.223 | **1.565** |

Table 5: Negative log-likelihood (NLL) of different fine-tuning strategies with simulated label noise. Noise is simulated using a high noise level setting $l = 0.5$.

to noise correction, MFFT presents an improvement in F1 score of $0.21$ ($+44\%$) and a reduction in ECE of $0.121$ ($-48\%$). Compared to Low-High, MFFT presents an improvement in F1 score of $0.019$ ($+2.8\%$) and a reduction in ECE of $0.141$ ($-51\%$).

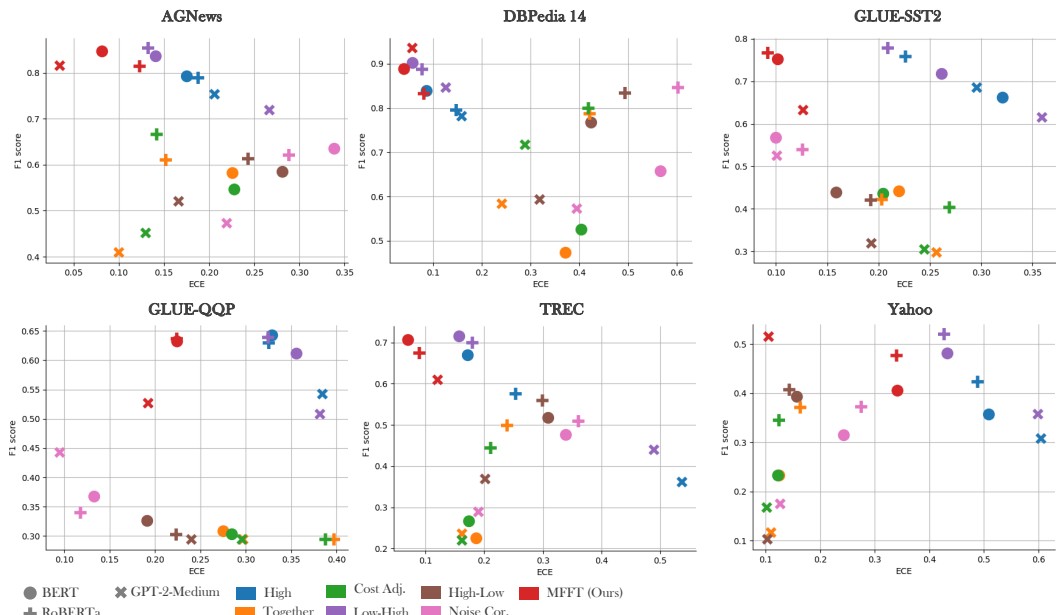

Figure 6: Macro F1-Score vs. expected calibration error (ECE) with high noise level $l = 0.5$. Favorable performance is in the top left corner of the graphs, where classification performance is high (high F1-score) and, simultaneously, calibration error is low (low ECE).

## C.3 SIMULATED EXPERIMENTS ABLATIONS

Using the TREC data set and RoBERTa as pre-trained language model, we vary different experimental settings to study performance at varying conditions. These include number of clean training examples constituting $\mathcal{D}_H$, noise level $l$ and noise bias $b$. Results are shown in figures 7, 8 and 9.

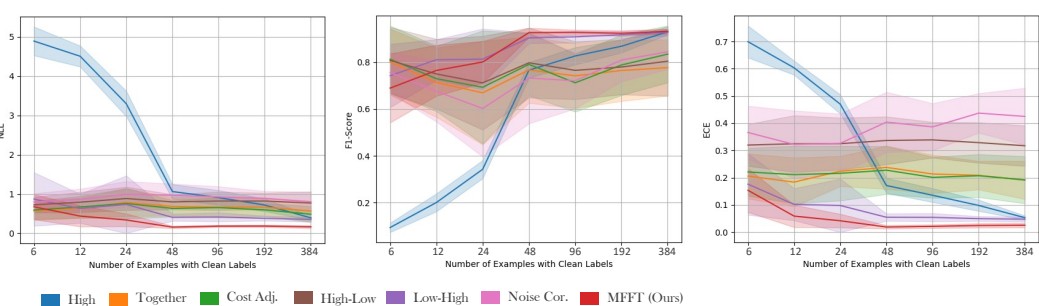

Figure 7: NLL, macro F1-score and ECE as a function of number of examples with clean labels available. Noise level is fixed at $l = 0.3$ and noise bias is fixed at $b = 0.3$

In figure 7, we observe that particularly High, Low-High and MFFT appreciaby improve as more clean data is made available for training. The extreme differences displayed by High are expected, as this is the baselines that uses solely clean data. MFFT is competitive or better than the best baseline (Low-High) for all metrics at all values of clean data size. F1-score is aligned with Low-High at all clean data budgets, while NLL and ECE are consistently better. This indicate that classification performance is generally comparable to Low-High (transfer learning approach), while providing superior calibration, independently of the amount of available clean data.

The improvement provided by MFFT is even more evident across different noise properties. In figures 8 and 9 MFFT performs comparably or better than all baselines across all three metrics. MFFT performance is also noticeably more robust to increasing noise level and bias, remaining stable as the noise level and bias in noisy data are increased.

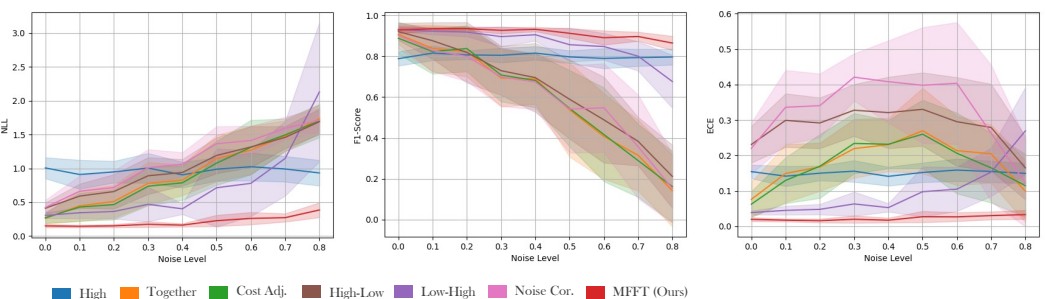

Figure 8: NLL, macro F1-score and ECE as a function of noise level applied to the noisy labels in the training set. Number of clean examples is fixed at $50$ and noise bias is fixed at $b = 0.3$

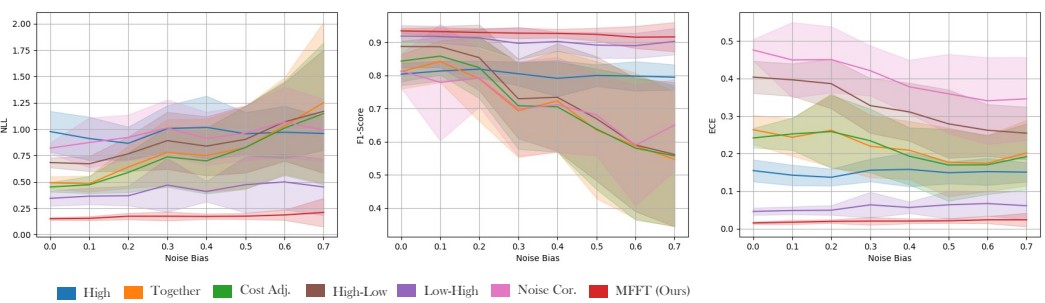

Figure 9: NLL, macro F1-score and ECE as a function of noise bias in the noise process used to simulate noisy labels in the training set. Number of clean examples is fixed at $50$ and noise level is fixed at $l = 0.3$

## C.4 ADDITIONAL EXPERIMENTS WITH REAL NOISE FORM LLM GENERATED DATA

We perform experiments with real noise from LLM generated data analogous to those presented in section 4.3, but with a smaller starting clean data set $\mathcal{D}_L$ of 50 examples. We show NLL results in table 6 and F1 vs. ECE in figure

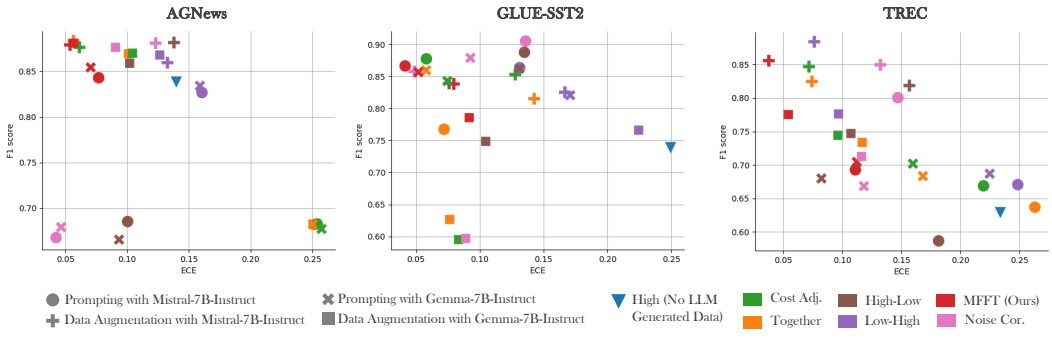

Figure 10: Macro F1-Score vs. expected calibration error (ECE) for RoBERTa fine-tuned on $50$ clean examples and $5,000$ examples generated with LLMs. Favorable performance is in the top left corner of the graphs, where classification performance is high (high F1-score) and, simultaneously, calibration error is low (low ECE).

Similarly to the results of section 4.3, MFFT results in the lowest or within statistical significance of the lowest NLL in all experiments and is better than any baseline in $58\%$ of cases. The F1 vs. ECE results of figure 10 also follow the trends observed in figure 4, with MFFT displaying consistently competitive classification performance and calibration. Across experiments, the most competitive baseline in classification performance (average F1) is Low-High, which MFFT outperforms by $0.013$

| Data Set | High | Together | Cost Adj. | Low-High | High-Low | Noise Cor. | MFFT |
|---|---|---|---|---|---|---|---|
| **Prompting with Mistral-7B-Instruct** | | | | | | | |
| AGNews | 1.084 | 1.483 | 1.501 | 1.252 | 0.845 | 0.608 | **0.509** |
| GLUE-SST2 | 1.667 | 0.437 | **0.339** | 1.111 | **0.376** | **0.349** | **0.336** |
| TREC | 1.633 | 1.262 | 1.124 | 1.772 | 1.177 | **0.692** | **0.654** |
| **Data Augmentation with Mistral-7B-Instruct** | | | | | | | |
| AGNews | 1.176 | 0.386 | 0.417 | 1.043 | 0.461 | 0.456 | **0.373** |
| GLUE-SST2 | 1.745 | 0.691 | 0.664 | 1.387 | **0.431** | **0.397** | **0.459** |
| TREC | 1.676 | 0.551 | 0.466 | 0.561 | 0.567 | 0.503 | **0.287** |
| **Prompting with Gemma-7B-Instruct** | | | | | | | |
| AGNews | 1.155 | 1.449 | 1.406 | 1.276 | 0.917 | 0.627 | **0.454** |
| GLUE-SST2 | 1.766 | 0.386 | 0.446 | 1.395 | **0.416** | **0.368** | **0.362** |
| TREC | 1.622 | 1.207 | 1.141 | 1.614 | 1.015 | 0.881 | **0.652** |
| **Data Augmentation with Gemma-7B-Instruct** | | | | | | | |
| AGNews | 1.154 | 0.567 | 0.591 | 0.961 | 0.498 | 0.501 | **0.373** |
| GLUE-SST2 | 1.641 | 0.654 | **0.601** | 1.924 | **0.541** | **0.598** | **0.570** |
| TREC | 1.672 | 0.643 | 0.579 | 0.729 | 0.533 | 0.590 | **0.381** |

Table 6: NLL of RoBERTa fine-tuned with different approaches on 50 clean labeled examples together with $5,000$ LLM generated examples. Values which are lowest or within statistical significance of the lowest over 5 experiment repeats are shown in bold.

in F1 score ($+1.55\%$) and 0.09 in ECE ($-56.4\%$). The most competitive baseline in calibration (average ECE) is Noise Correction, which MFFT outperforms by 0.038 in F1 score ($+4.89\%$) and 0.029 in ECE ($-29.2\%$).

