# OpenReview forum: "Multi-Fidelity Fine-Tuning of Pre-Trained Language Models"
_ICLR.cc/2025/Conference — ICLR 2025 Conference Withdrawn Submission_

### Official Review · Reviewer_CDy4 · 2024-10-30

**Soundness:** 2
**Presentation:** 3
**Contribution:** 2
**Rating:** 3
**Confidence:** 3

**Summary:**

The authors present a training framework designed to handle mixed-quality training data. Specifically, they propose training two separate models: one on abundant but noisy data, and another on limited high-quality data. The framework then generates pseudo-labels based on the predictions from the model with higher confidence, using these labels to train on the entire dataset. Applied to classification tasks with language models, this MFFT framework demonstrated superior performance compared to other training techniques.

**Strengths:**

The paper is easy to understand and the authors conduct abundant experiments comparing with many baseline models and datasets.

**Weaknesses:**

Major Concerns:

- The authors propose a general mixed-quality training framework not specifically tailored to language models, and the pseudo-labeling strategy—resembling a voting technique—lacks novelty.
- The experimental results are unconvincing. The authors employ a complex training technique (ensemble) and significantly more training epochs compared to baseline methods, making the comparisons less fair. Given the availability of more sophisticated training approaches, such as teacher-student models, the authors only benchmark against relatively simple baseline methods.

Minor Issues:

- Table 3 is unclear; the authors should explain how they calculated NLL and derived the confidence intervals.

**Questions:**

From the weakness section

---

> ### Author Response · Authors · 2024-11-22
> **Clarification on Proposed Baselines**
>
> “Given the availability of more sophisticated training approaches, such as teacher-student models, the authors only benchmark against relatively simple baseline methods”
>
> We would like to ask the author some clarification about this comment. We have compared to all applicable methods we could find in the literature to fine-tune pre-trained model with two data sets of different quality. Of course, we may have missed some. Can the reviewer elaborate on how teacher-student models can be used to fine-tune in this multi-fidelity scenarios? Could the reviewer provide any references?

---

### Official Review · Reviewer_dDG5 · 2024-11-02

**Soundness:** 2
**Presentation:** 2
**Contribution:** 2
**Rating:** 3
**Confidence:** 3

**Summary:**

This paper presents MFFT to tackle the setting where only a small amount of high-fidelity labeled data and a relatively large pool of low-fidelity (noisy) data are available. The proposed MFFT first trains two ensembles based on noisy data and all available data (with the noise to real order). Then a final classification model is distilled from the two ensembles with a soft labeling criterion based on the expected log-likelihood, or uncertainty estimation of the two deep ensembles. The empirical result shows that the proposed method outperforms several vanilla baselines on text classification tasks under both synthetic and LLM-generated noise settings.

**Strengths:**

* Most of the paper is easy to follow. The related works and background context are well discussed.
* The problem setting and method of this work is well-motivated.
* The final performance is nice compared to the baselines under the text classification tasks.

**Weaknesses:**

* There should be more justification than provided for why the average log-likelihood of low-fidelity models over $(x_H, y_H)$, i.e., $\log p_L(y_H | x_H)$ is a robust estimation of the expected log-likelihoods, which is a constant solely based on the performance of the low-fidelity model on $(x_H, y_H)$. Can you further justify why "the expected log-likelihood of the low fidelity model to be approximately constant with respect to $x$" is a valid assumption, either mathematically or empirically?
* The comparison against other baselines is somewhat unfair since the proposed MFFT requires maintaining two deep ensembles and an extra soft labeling process for distillation, which requires more computation than the baselines. Also, it is known that using deep ensemble and label smoothing could improve classification performance and calibration. I think there should be more ablation studies against vanilla deep ensemble and label smoothing to demonstrate the effectiveness of MFFT.
* Although the multi-fidelity setting is very common in practice, this work mainly considers the traditional text classification tasks. I would like to count the limited scope as a weakness of this work, which can be improved by conducting additional experiments, or at least discussing the applicability on more general LLM fine-tuning settings for the proposed method.

### Minor
* There seems to be a typo on the LHS of Eqn. (5).

**Questions:**

* How much extra computation is needed for MFFT compared to the baselines?
* Can you provide some thoughts on tackling more general multi-fidelity settings beyond text classification tasks?
* Have you considered any other thresholding criterion for soft labeling?

---

> ### Author Response · Authors · 2024-11-22
> **Answers to Reviewer's Questions**
>
> - MFFT does require more training compute. Roughly, if the number of ensemble is N, the compute will be N+1 time standard fine-tuning. We use N=5 for our experiments.
>
> - The method could be applied to generative LLMs too on a token-by-token level. This would bring its own practical challenges, especially in evaluating and comparing uncertainty of the ensembles in our objective function; every new token, we would need to infer the probability with every member of the ensemble, for high and low fidelity, and compute our objective function, sample from the resulting infer distribution to get the next token, and then repeat.
>
> - Yes, we have experimented with other Bayesian objectives, such as mutual information between high-fidelity model and new samples. The current mechanism comparing entropy to mean historical log likelihood of the low fidelity model performed best.

---

### Official Review · Reviewer_6JHo · 2024-11-03

**Soundness:** 1
**Presentation:** 1
**Contribution:** 1
**Rating:** 3
**Confidence:** 5

**Summary:**

This paper focuses on the setting where high-quality data is limited for LLM fine-tuning. Given this, the authors consider leveraging low-quality data (which are often more sufficient) to augment the training set. The authors propose to separately train two set of models on high and low quality data, and then annotate the low-quality data using the two set of models with a designed rule. The authors perform some small-scale experiments on small language models to explore the effectiveness of the method.

**Strengths:**

I do not find any strengths of this paper, except for the setting it explores. It is very important to study how to better fine-tune the LLMs when high-quality data is limited

**Weaknesses:**

This paper does not make any sense at all. Despite the problem the authors studied is interesting and meaningful, the proposed method, the derivations, and the judgements are completely wrong. The authors include a lot of equations and so-called "proofs" or "derivations". However, these derivations are completely wrong and are mainly used to mystify the readers. The experiment sections are also poorly organized. Below are the detailed points for the weakness.

**First, completely wrong derivations.** The methodology section (Section 3) is basically wrong. Let's analyze each equation one-by-one. The authors first state the negative log-likelihood loss for training in Equation (1) for classification tasks. Then the authors leverage the idea of Deep Ensembles with slight revision to train a set of models on both low and high fidelity data. These parts are resonable. However, starting from `Equation (3)`, everything changes.

In equation (3) which is the core of the authors "theory", the authros discuss how to choose from the two models (trained and ensembled on high/low fidelity data) given a **testing input**. The "objective function" is:

$$\alpha ^* = \underset{\alpha}{\mathrm{argmax}} \mathbb{E} _{p(y|x)} \mathrm{log}p _{H}(y|x) ^{1-\alpha}\mathrm{log}p _{L}(y|x)^{\alpha}$$

This is quite funny. The authors claim that $x$ is a **testing input**, but the expectation is taken over $p(y|x)$, which follows the groud-truth distribution and is never available. This is not even a valid objective function -- you only have **one testing example** in the objective function, what you are doing to take the expectation over inputs??? Do not tell me your expectation is taking over the class labels. That is not a expectation but just a summation over $p(y_1|x), ..., p(y_n|x)$ where $n$ is the number of classes. (By the way, this aim of this equation is nothing but comparing the prediction probability of the ground-truth label $y$ for testing input $x$ using the two models and choose the one with higher probability.)

No worries, this objective function is invalid, but the authors claim we can find a way to estimate it and thus solving $\alpha$ for that input $x$. The first thing the authors did is to estimate the first term, $\mathbb{E} _{p(y|x)} \mathrm{log}p _{L}(y|x) ^{1-\alpha}$, which is believed by the authors to be easier. Particularly, the authors claim that:

$$\mathbb{E} _{p(y|x)} \mathrm{log}p _{H}(y|x) \approx \mathbb{E} _{p _{H}(y|x)} \mathrm{log}p _{L}(y|x),$$

in equation (8) in appendix A.1.

Again, let's ignore the issue with expectation as there should not be expectation at all. The explanation from the authors is that "the ensemble of each model trained on high-fiedelity data, which is $p_{H}(y|x)$, are un-biased estimation of the true mapping $p(y|x)$". I would like to recommend the authors to re-consider their logic here. There basic assumption is that the amoung of high-fiedality data is limited, but they state that they can estimate the true distribution using limited high-fiedality data already. In that case, why do not directly use your ensembled model trained on high-fiedality data? The correct assumption should be that due to the limited high-fiedality data, we cannot get an accurate estimation of $p(y|x)$. If you already can get that, then why we still need to other things?

As long as we slightly think about the logic, we can find **the first line of Equqation (8) is completely wrong**.

No worries, let's continue with the authors' great math. Since we have discussed the first time in their objective function, we learn from this paper to estimate the second term. Again, no expectation is needed here.

$$\mathbb{E} _{p(y|x)} \mathrm{log}p _{L}(y|x) = \frac{1}{M}\sum _{i}^{M}\mathrm{log}p _{L}(y _{H,i}|x _{H,i})$$

If we remove the expectation (which should not be there) in the left-hand side (LHS), we get

$$\mathrm{log}p _{L}(y|x) = \frac{1}{M}\sum _{i}^{M}\mathrm{log}p _{L}(y _{H,i}|x _{H,i})$$

This is even more ridiculous. The author trying to tell us that, **for any given testing example, the model's prediction should be equal to the average prediction probability on another set of examples that are completely different and not related to the testing example.** This might be one of the most impressive conclusion I have ever read.

The reason, however, is there should be no expectations. The authors wrongly add the expectation when considering a testing example, which leads to the wrong conclusion. Of course, if we consider two extremely large set of examples (one for the LHS and another for the high-fidelity data which is the RHS), then the original equation may hold. But this is not the case.

In summary, the estimation of the objective function is completely wrong. Thus, the solution to it is also wrong (Equation 6).

**Second, although the derivations are wrong, the equation (6) is actually somehow interpretable.** It basically want to compare the uncertainty of the two models trained on low/high fidelity data and choose the prediction with lower uncertainty. However, due to the wrong derivation, the two terms are actually not meaningful. One is about the entropy measuring the predictive uncertainty of the model trained on high-fidelity data. Another one, however, is the average log probability of the model trained on low-fidelity data on a particular dataset. Such a comparison is counter-intuitive. These two terms have different meanings, but the authors do not consider this and check their derivations.

I am opening to any discussions and would like to apologize to the authors sincerely if my understanding is wrong.

**Questions:**

See the weakness above.

---

> ### Author Response · Authors · 2024-11-22
> **Review Response**
>
> The reviewer is making several core mistakes. Detail responses to their comments below:
>
> COMMENT: “This is quite funny. The authors claim that x is a testing input, but the expectation is taken over p(y|x), which follows the groud-truth distribution and is never available. This is not even a valid objective function — you only have one testing example in the objective function, what you are doing to take the expectation over inputs??? Do not tell me your expectation is taking over the class labels. That is not an expectation but just a summation over p(y_1|x), p(y_2|x)..., p(y_n|x). is the number of classes.”
>
> RESPONSE: The expectation over the outputs, given a single input is absolutely valid. It is a core notion in probabilistic machine learning at the basis of metrics such as entropy and KL divergence. In the case of categorical distributions, this is the sum of the quantity inside the expectation for each class, weighted by the respective class probabilities. Our ideal objective is indeed for a new test input and we do not have the true process p(y|x), hence the approximations that follow. It is important to state the ideal objective, even though we cannot compute it, in order to understand what the final objective, which we can compute, is targeting and with what approximations.
>
> COMMENT: “Again, let's ignore the issue with expectation as there should not be expectation at all. The explanation from the authors is that "the ensemble of each model trained on high-fiedelity data, which is p_H(y|x), are un-biased estimation of the true mapping p(y|x)". I would like to recommend the authors to re-consider their logic here. There basic assumption is that the amoung of high-fiedality data is limited, but they state that they can estimate the true distribution using limited high-fiedality data already."
>
> RESPONSE: Firstly, the reviewer is making many notation errors in the equations they are quoting from our paper, but we think we understand what they are saying in this comment. The reviewer is confusing estimating the true distribution with estimating the expectation of model log likelihood under the true distribution. These are very different, but we can see how this may be confusing under the wrong assertion by the reviewer that expectations over outputs should not exist. Let’s make a simple example:
>
> Consider the binary case where p(y=1|x_i) = 1 and p_H(y=1|x_i) = 0.5. Now, these distributions are clearly very different, but the expectation of model log likelihood under the true distribution and under the model itself are the same:
>
> $$
> \mathbb{E}_{p(y|x_i)} \log p_H(y|x_i) = 1 \times \log(0.5) + 0 \times \log(0.5) = \log(0.5)
> $$
>
> $$
> \mathbb{E}_{p_H(y|x_i)} \log p_H(y|x_i) = 0.5 \times \log(0.5) + 0.5 \times \log(0.5) = \log(0.5)$
> $$
>
> In general, for the negative entropy of the model to be close to the expectation of its log likelihood under the true output distribution, it requires the model to be well calibrated, but NOT the same as the true output distribution. This is the assumption we make and what we achieve by building p_H(y|x) with an ensemble.
>
> COMMENT: “If we remove the expectation (which should not be there) in the left-hand side (LHS), we get... ...This is even more ridiculous. The author trying to tell us that, for any given testing example, the model's prediction should be equal to the average prediction probability on another set of examples that are completely different and not related to the testing example. This might be one of the most impressive conclusion I have ever read.”
>
> RESPONSE: Again, this mistake made by the reviewer derives from the erroneous notion that expectations over outputs are generally an invalid concept. Equation 5 in the paper DOES NOT state that the model output distribution p_L(y|x) is always approximately equal to its average over the inputs x_H, it states that its expected log likelihood under the true process is, which is different. In simple terms, this coarse approximation is saying that, when we run the model P_L(y|x_i) on a new input x_i, we expect its log likelihood to be approximately the historical average over previous inputs x_H. Again, this DOES NOT imply that p_L(y|x_i) is approximately equal to the average of the predictions p_L(y_H|x_H).
>
> COMMENT: “However, due to the wrong derivation, the two terms are actually not meaningful. One is about the entropy measuring the predictive uncertainty of the model trained on high-fidelity data. Another one, however, is the average log probability of the model trained on low-fidelity data on a particular dataset.”
>
> RESPONSE: The two terms are absolutely comparable; they are both estimates of the expected log likelihood of the two models. Entropy is exactly the negative of the expectation of log likelihood, under the model itself. The second term is also an expectation of the log likelihood, but based instead on historical performance over ground-truth points [x_H, y_H].

---

### Official Review · Reviewer_oitd · 2024-11-03

**Soundness:** 3
**Presentation:** 4
**Contribution:** 2
**Rating:** 5
**Confidence:** 2

**Summary:**

In this work, the authors propose Multi-Fidenity Fine Tuning (MFFT), a method for finetuning pretrained language models on both low and high quality data. MFFT works by first finetuning the model on the low quality data. Another ensemble of models is then derived by further finetuning on the high quality data. The low quality data model and high quality data model are then used to re-label the low quality data. Each model makes a prediction on each low quality data point, and the prediction which maximizes the expected log likelihood of the data is selected as the label. The authors provide evaluations in both the setting where the low-quality data is derived through a synthetic noising process and the setting where the low quality data is data labled by an LLM.

**Strengths:**

* The methodology in the paper is well-explained, and the writing throughout the paper makes the results and takeaways very clear.

* MFFT is evaluated on multiple benchmarks for both synthetic and real data and appears to consistently outperform the baselines considered.

**Weaknesses:**

* It seems like MFFT requires significantly more training compute due to the multiple rounds of training and training of an ensemble network. It seems important to have a compute matched experiment across all multi-fidelity methodologies to determine which method most effectively uses compute.

**Questions:**

* I think equation 5 should have $\log p_L(y | x)$ instead of $\log p_H(y | x)$ on the LHS.

* Did you perform any experiments to see how the different finetuning methods scale with different ratios of high-quality to low-quality data?

---

> ### Author Response · Authors · 2024-11-22
> **Answers to Reviewer's questions**
>
> - Thank you for spotting the mistake; indeed this is the case.
> - Yes; supplementary C3 contains ablations over the number of high-fidelity samples used, as well as noise level and noise bias in the low-fidelity data.

---

### Author Response · Authors · 2024-11-22
**General Response**

We thank the reviewers for the useful and thoughtful feedback. The main comments are concerned with the experimental comparison to compute-matched baselines and other ensemble methods, as the proposed technique does require more training compute than existing approaches in the literature, which we tested. We value this feedback very much and we will perform these experiments for the next iteration of the paper. We provide below a detailed response to reviewer 6JHo, as their review contains major misunderstanding. For completeness, we also answer direct questions by the other reviewers.

---

### Note · Authors · 2024-12-16

**Comment:**

We thank again all reviewers for the constructive feedback. We will extend experiments and clarify the theory as suggested and resubmit to a later venue.

**Withdrawal Confirmation:**

I have read and agree with the venue's withdrawal policy on behalf of myself and my co-authors.